# Do you take off your mask correctly? A survey during COVID-19 pandemic in Ningbo, China

**Jingjing Ma**[1]*, **Yiqing Zhang**[1], **Shunshun Lu**[2], **Shiyong Chen**[2], **Yuezheng Rong**[1], **Zhengzheng Wang**[1]

1 Department of Nursing, Ningbo Medical Center LiHuili Hospital, Ningbo, China, 2 Department of Infection, Ningbo Medical Center LiHuili Hospital, Ningbo, China

* shelleymjj2004@hotmail.com

## Abstract

Guidelines and recommendations from public health authorities related to face masks have been essential for containing the COVID-19 pandemic. A cross-sectional survey was conducted in Ningbo City, China, from April 8 to 12, 2022. We assessed the behavioral differences and correlates of mask usage, primarily mask-removal. We examined public mask-wearing behavior during on-site COVID-19 nucleic acid detection. The survey instrument was developed based on the guidelines issued by the World Health Organization and consisted of demographics, mask-wearing knowledge, and behavior. We analyzed data from 1180 participants; 73.2% demonstrated good mask-wearing knowledge. However, regarding mask-wearing behavior, only 53.7% knew the correct way to remove a mask; 70.3% maintained hand hygiene after touching the outside. Binary logistic regression analyses revealed that health prevention knowledge and free mask distribution were positively associated with two types of mask-wearing behaviors. Most participants used masks during the COVID-19 pandemic; however, mask-removal and hand hygiene were neglected when touching the outside of the mask. More attention must be paid to mask-removal and hand hygiene details. Local health authorities should consider introducing the free distribution of masks.

## Introduction

Coronavirus has emerged as a global health threat due to its accelerated geographic spread over the last two decades [1]. The WHO recommended face coverings or masks in public to manage the crisis. Many countries and regions recommend face masks to control the spread of the severe acute respiratory syndrome coronavirus 2 (SARS-CoV-2), the disease that causes COVID-19 [2–5]. Risk assessment studies using population transmission models showed that population-wide mask-wearing could slow the spread of influenza pandemics [6]. Laboratory studies suggested that proper mask using can reduce the spread of aerosol and respiratory droplets of coronaviruses, influenza viruses, and rhinoviruses [7].

The use of masks by the public might be one of the most effective strategies to reduce the spread of COVID-19, as recommended by the National Health Commission of China in

had no role in study design, data collection and analysis, decision to publish, or preparation of the manuscript.

**Competing interests:** The authors have declared that no competing interests exist.

guidelines issued on March 18, 2019 [8]. The Chinese government has strongly advocated using masks in public places as a source control method. The Chinese generally support the usage of face masks in a public setting as a supplement to hand hygiene and social distancing to slow down or contain the exponential growth of the epidemic [9]. During the uncertain time of a global crisis, one of the most efficient mitigation methods is face coverings or masks, which can play an essential role in COVID-19 disease control [10].

Nevertheless, experts and government health authorities warn that incorrect mask use may increase the risk of infection [11]. Previous studies showed that personal beliefs, individual attitudes, and sociodemographic factors predicted mask-wearing during other health emergencies [12]. Other studies showed that residents might not wash their hands, change masks frequently, or dispose of used masks properly, simply by recording mask usage rates [13–15]. We were curious about the rate of correct mask usage on a general population scale, mainly whether the rate was related to differences in mask-wearing behavior.

Currently, the epidemic in China is characterized by local sporadic and large-scale aggregation. The National Health and Health Commission guides all localities to promote nucleic acid testing (2019-nCoV) to identify high-risk individuals. From April 8 to 12, 2022, Zhejiang Ningbo City opened 276 nucleic acid sampling locations and performed large-scale nucleic acid testing.

COVID-19 can spread through droplets and contact infected persons directly [16]. It survives on surfaces for up to 72 hours. Contact with contaminated surfaces with subsequent touching of the face is another possible transmission source [17]. The areas with high population densities may be more at risk, especially in large-scale nucleic acid testing sites. Therefore, we investigated mask-wearing behavior patterns during large-scale nucleic acid testing to identify the factors that might explain differences in mask-wearing behavior.

## Materials and methods

### Participants

This cross-sectional survey was conducted from April 8 to 12, 2022. Due to new coronavirus infections in Ningbo, the corresponding area carried out nucleic acid testing for all staff. We conduct an investigation on-site during this period, relying on the authors as medical staff, community workers, volunteers, and a nucleic acid sampler. Inclusion criteria: people over 16 agreed to participate and have the ability to read and write in Chinese. There were no other exclusion criteria. Participants scanned a QR code of the questionnaire while waiting for nucleic acid sampling. The data collection was performed predominantly online.

### Ethical considerations

The study survey was reviewed and approved by the Institutional Review Board of Li Huili Hospital of Ningbo Medical Center (No: KYSB2022SL246). All procedures were performed the relevant guidelines and regulations. A brief introduction explained the background and purpose of the survey. The questionnaire was scanned using the QR code at the site. Informed consent was obtained from each participant, and they received on-site instruction as they completed the survey. Minors aged 16–18 have obtained the consent of their parents or guardians. We carried out all procedures according to relevant regulations and guidelines.

### Instruments

The survey instrument was developed based on the guidelines issued by the WHO [18] and community management of COVID-19 by the National Health Commission of the People's

**Table 1. Questionnaire of health prevention knowledge and behavior of mask-wearing toward COVID-19.**

| Health prevention knowledge of mask-wearing (correct rate, % of the total sample) | Options |
|---|---|
| 1. Do you know social distancing? (90.8) | Yes/no |
| 2. Do you know relevant knowledge about wearing masks? (94.3) | Yes/no |
| 3. Which type of mask do you choose? (92.3) | Disposable medical mask/Cloth mask |
| 4. Do you know the correct way to wear a mask? (85.0) | Yes/no |
| 5. Do you clean your hands before wearing a mask? (79.8) | Never/occasionally/often/always |
| 6. Will metal bars be facing up when the mask is used? (89.7) | Never/occasionally/often/always |
| 7. When using the mask, should the dark color face out? (87.3) | Never/occasionally/often/always |
| 8. Do you wear a mask covering your nose, mouth, and chin? (81.5) | Never/occasionally/often/always |
| 9. When you use a mask, do you hang the mask on your chin? (79.2) | Never/occasionally/often/always |
| 10. Do you expose your mouth or nose when using a mask? (82.6) | Never/occasionally/often/always |
| 11. Do you touch the outside of the mask while using it? (82.2) | Never/occasionally/often/always |
| 12. Which side of the mask do your hands touch when taking off the mask? (69.5) | Outside up/outside down/ Inside down/from the thin straps |
| Health prevention behavior of mask-wearing (correct rate, % of the total sample) | Options |
| 13. During nucleic acid sampling, how do you remove your mask? (53.7) | From the thin straps/Inside down/Outside up/ outside down |
| 14. Do you perform hand hygiene if you touch the outside of the mask? (70.3) | Never/occasionally/often/always |

Republic of China [19, 20]. The questionnaire was modified based on their comments. The final version of the questionnaire was entitled "Questionnaire on knowledge and behavior of masks during COVID-19 nucleic acid sampling". The questionnaire was created and distributed in the Chinese language. It consisted of three parts: (1) Demographics: age, gender, education, occupation, registered residence, control area and the unit providing masks; (2) Health prevention knowledge of mask-wearing, which consisted of 12 items: keeping a safe distance, mask selection, mask-wearing knowledge, the timing of hand hygiene, mask compliance, face mask usage, and hand washing; (3) Health prevention behavior of mask-wearing, consisting of two items: the way of taking off your mask and perform hand hygiene. The Likert scale assessed the questions with four points (never, occasionally, often, every time) (Table 1). We used binary logistic regression to analyze risk factors for mask-wearing and hand hygiene behaviors, and the expected sample size was 500.

## Outcome variable definition

The primary outcome was the mask-removal method during nucleic acid sampling, based on the survey question: "from the thin straps of masks, inside down, outside up, and outside down." The correct way of taking off masks was from the thin strap. The wrong ways of taking off masks include finger going into the inside of the mask and taking off masks inside down, touching the mask's surface, lifting the mask up, touching the mask's surface, and lifting the mask down. We defined the method of mask-removal as a binary variable: 1 if the respondent reported taking off the mask from the thin straps and 0 otherwise.

The secondary outcome was that if you touch the outside of the mask, do you use hand hygiene? Based on the survey question (never; occasionally; often; always). We defined hand hygiene as a binary variable: 1 if the respondent reported often or always, and 0 otherwise.

## Dependent and independent variables

The dependent variables were health prevention behavior of mask-wearing measured, assessed using the following questions: During the nucleic acid sampling process, how do you take off your mask? Do you perform hand hygiene if you touch the outside of the mask? The independent variables included in this study are displayed in Table 2. Demographic variables were gender, age, education, residence, professional, free mask distribution, control area, and health prevention knowledge.

## Statistical analysis

The data were analyzed using SPSS™ for Windows, version 23.0. Parallel mediation effects analysis was conducted using the Hayes approach (2013) and PROCESS Macro by performing a bias-corrected bootstrap procedure. We used health prevention knowledge as descriptive statistics. Other data were categorical variables expressed as frequencies or percentages. Incorrect and correct responses were scored 0 and 1, respectively. Items 1, 2, and 4 were "yes" or "no." Items 5, 6, 7, and 8 that were answered "occasionally" and "never" were defined as incorrect, while "always" and "often" was defined as correct. Items 9, 10 and 11 were the opposite. For item 3, responses of "disposable medical mask" were defined as correct. For item 12, responses

**Table 2. Demographic characteristics of the study population participants of the Correlation of hand hygiene after touching the outside of the mask and taking off the mask N = 1180, number of cases(percentage).**

| Category | n% | Hand hygiene | | $\chi 2$ | p | Take off mask | | $\chi 2$ | p |
|---|---|---|---|---|---|---|---|---|---|
| | | Good Rate | Poor Rate | | | Good Rate | Poor Rate | | |
| **Gender** | | | | 4.36 | 0.04 | | | 2.47 | 0.11 |
| Female | 675 (57.2) | 491 (59.2) | 184 (52.6) | | | 376(59.3) | 299(54.8) | | |
| Male | 505 (42.8) | 339 (40.8) | 166 (47.4) | | | 258(40.7) | 247(45.2) | | |
| **Age** | | | | 2.43 | 0.29 | | | 5.49 | 0.06 |
| <35 | 548 (46.4) | 375 (45.2) | 173 (49.4) | | | 279(44.0) | 269(49.2) | | |
| 35–65 | 584 (49.5) | 418 (50.4) | 166 (47.5) | | | 333(52.5) | 251(46.0) | | |
| >65 | 48 (4.1) | 37 (4.4) | 11 (3.1) | | | 22(3.5) | 26(4.8) | | |
| **Education** | | | | 14.51 | <0.01 | | | 16.84 | 0.01 |
| Middle school or below | 16 (14.1) | 137 (16.5) | 29 (8.3) | | | 100(15.8) | 66(12.1) | | |
| High school | 169(14.3) | 111 (13.4) | 58 (16.6) | | | 70(11.0) | 99(18.1) | | |
| College | 342(29.0) | 234 (28.2) | 108 (30.9) | | | 175(27.6) | 167(30.6) | | |
| bachelor degree | 503(42.6) | 348 (41.9) | 155 (44.3) | | | 289(45.6) | 214(39.2) | | |
| **City/Countryside** | | | | 4.47 | 0.03 | | | 2.98 | 0.08 |
| City | 649(55) | 440 (53) | 209 (59.7) | | | 334(52.7) | 315(57.7) | | |
| Countryside | 531(45) | 390 (47) | 141 (40.3) | | | 300(47.3) | 231(42.3) | | |
| **Medical staff** | | | | 0.93 | 0.33 | | | 0.07 | 0.79 |
| Yes | 159(13.5) | 117 (14.1) | 42 (12.0) | | | 87 (13.7) | 72(13.2) | | |
| No | 1021(86.5) | 713 (85.9) | 308 (88.0) | | | 547(86.3) | 474(86.8) | | |
| **Free masks distribution** | | | | 25.29 | <0.01 | | | 6.18 | 0.01 |
| Yes | 860(72.9) | 640 (77.1) | 220 (62.9) | | | 481(75.9) | 379(69.4) | | |
| No | 320(27.1) | 190 (22.9) | 130 (37.1) | | | 153(24.1) | 167(30.6) | | |
| **Control area** | | | | 0.10 | 0.75 | | | 20.01 | <0.01 |
| Yes | 447(37.9) | 312 (37.6) | 135 (38.6) | | | 203(32.0) | 244(44.7) | | |
| No | 733(62.1) | 518 (62.4) | 215 (61.4) | | | 431(68.0) | 302(55.3) | | |
| **Good knowledge** | | | | 44.35 | <0.01 | | | 173.1 | <0.01 |
| Yes | 864(73.2) | 654 (78.8) | 210 (60.0) | | | 564(89.0) | 300(54.9) | | |
| No | 316(26.8) | 176 (21.2) | 140 (40.0) | | | 70(11.0) | 246(45.1) | | |

of "from the thin straps" were defined as correct. There were 12 items (Table 1). According to the final score (total of 12 points), ≥ 10 points were regarded as "good," and < 10 points were regarded as "poor." Frequencies of health prevention knowledge answers and behavior of different persons were described, and data are presented as N (%). Chi-square tests were performed for behaviors of different populations according to demographic characteristics. One-way analysis of variance was performed with demographic characteristics and health prevention knowledge ($P < 0.05$) as independent variables and health prevention behaviors as dependent variables (items 13 and 14). The independent variables were included based on a literature review and our research hypothesis. Finally, binary logistic regression analyses were used to evaluate individual characteristics and health prevention knowledge levels (independent variables) were related to health prevention behaviors (dependent variable). Odds ratios (ORs) and their 95% confidence intervals (CIs) were used to quantify the associations between variables and behaviors. The statistical significance level was set at $p < 0.05$ (two-sided).

## Results and discussion

There were 1180 people participants who completed the survey with no missed cases. Women accounted for 57.2%, and participants aged 35–65 accounted for 49.5%. Those with medical

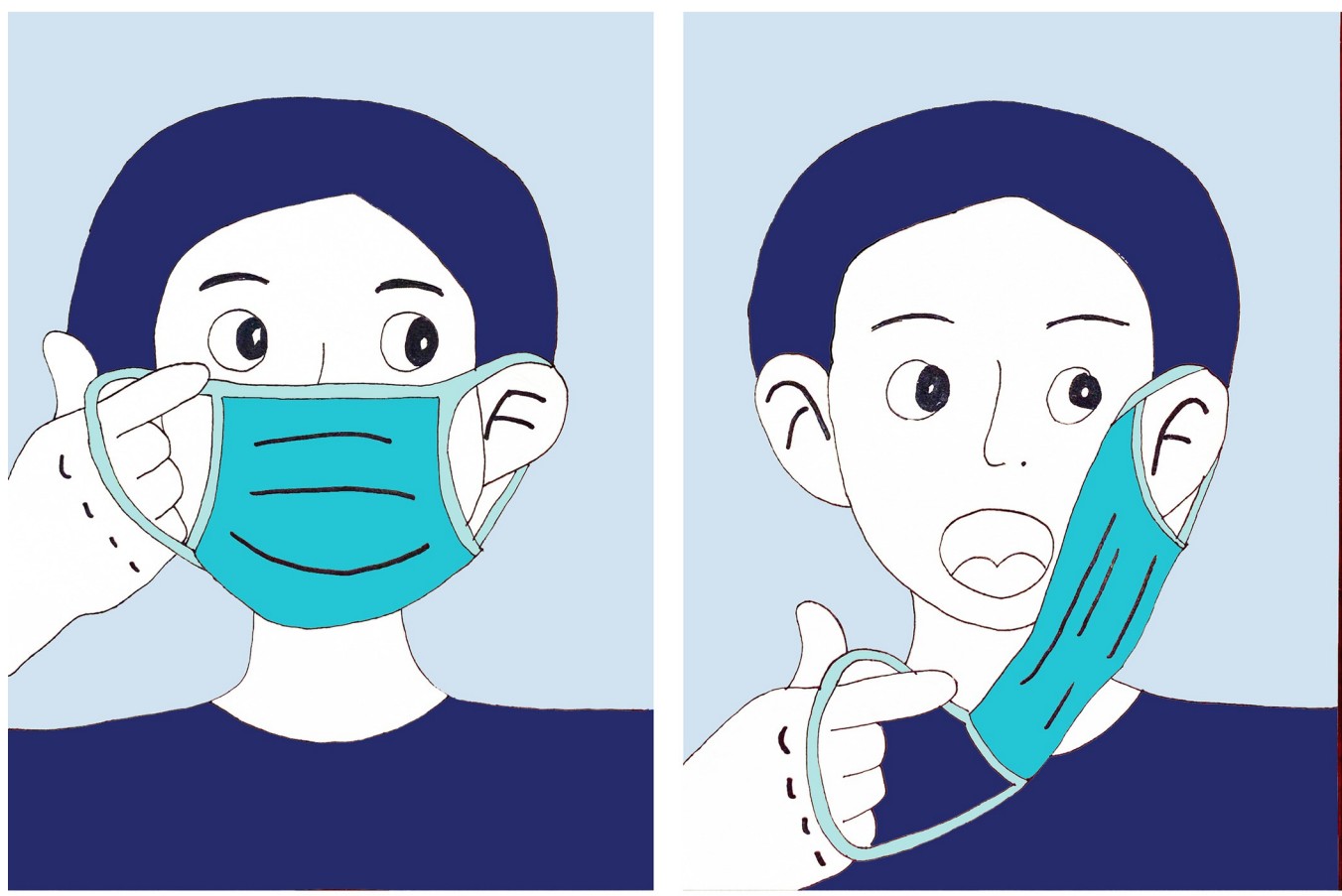

a

**Fig 1. It shows the correct way of removing masks from the thin strap.** This accounted for 53.7% of participants.

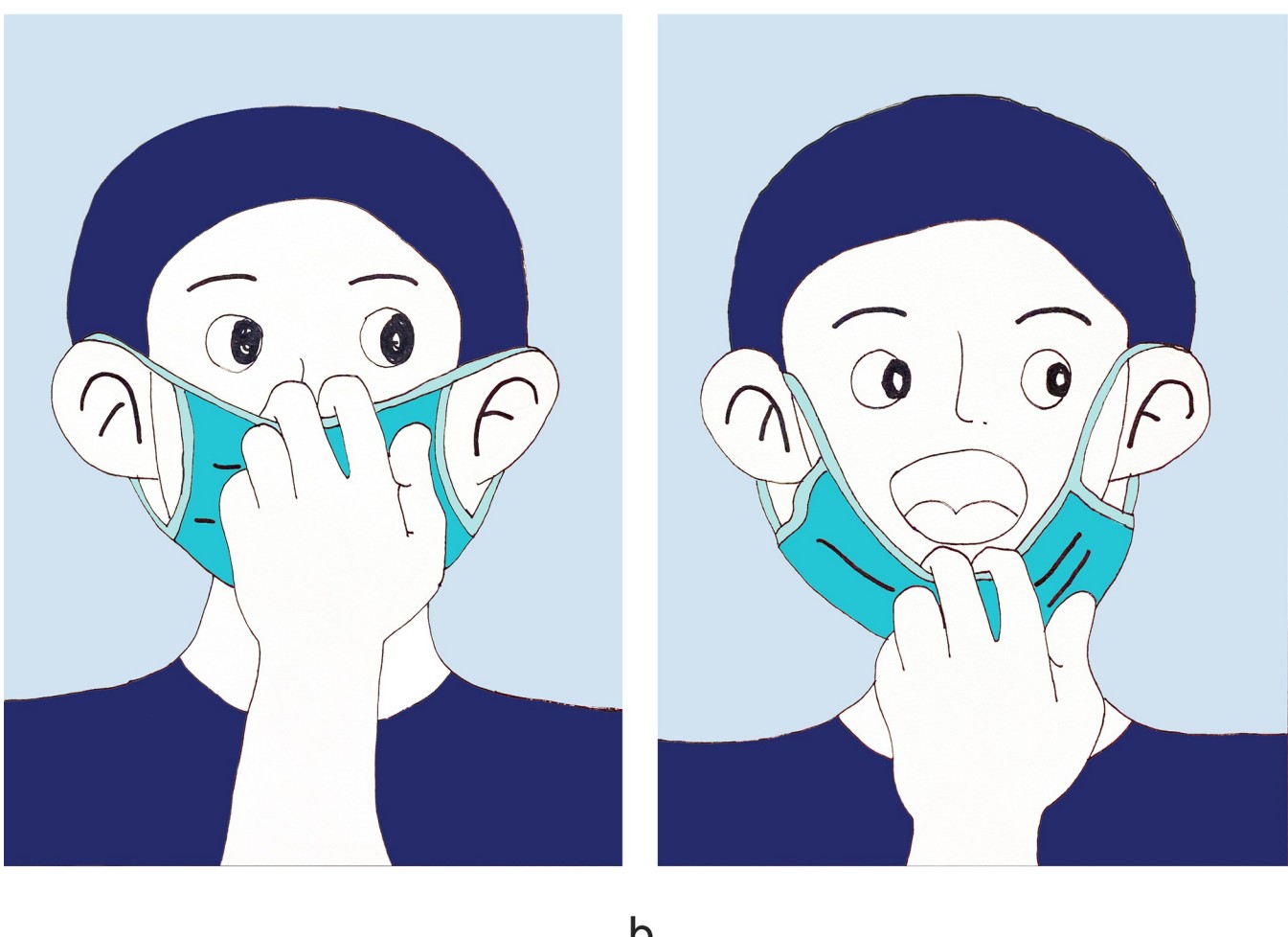

b

**Fig 2. It shows the wrong way of taking off masks.** The finger goes into the inside of the mask and removes the mask inside down. This accounted for 13.8% of the participants.

staff accounted for 13.5%. Regarding health prevention knowledge, 90.8% of the participants knew about social distancing, and 92.3% chose disposable medical masks (Table 1). 73.2% of the participants demonstrated good knowledge of face mask use (Table 2). The primary outcome was the mask-removal method during nucleic acid sampling. Regarding the survey, only 53.7% of participants knew the correct way to remove a mask (Fig 1); 13.8% of participants touch the inside of the mask and took off masks inside down (Fig 2); 3.7% of participants touch the surface of the mask, lift the mask up to expose the mouth (Fig 3); 28.8% of participants touch the surface of the mask, lift the mask down then expose the nose and mouth (Fig 4). Regarding hand hygiene, 70.3% used hand hygiene when touching the outside of the mask (Table 1). We performed a univariate analysis of the study population and found that gender, education, urban area residence, eceipt of free masks and health prevention knowledge were significantly related to hand hygiene when touching the outside of the mask (P<0.05) (Table 2). Education, receipt of free masks, control area and health prevention knowledge are significantly related to correct mask-removal during nucleic acid sampling (P<0.05). Binary logistic regression analysis identified health prevention knowledge and free face mask distribution were predictors of mask-wearing behaviors(Tables 3 and 4).

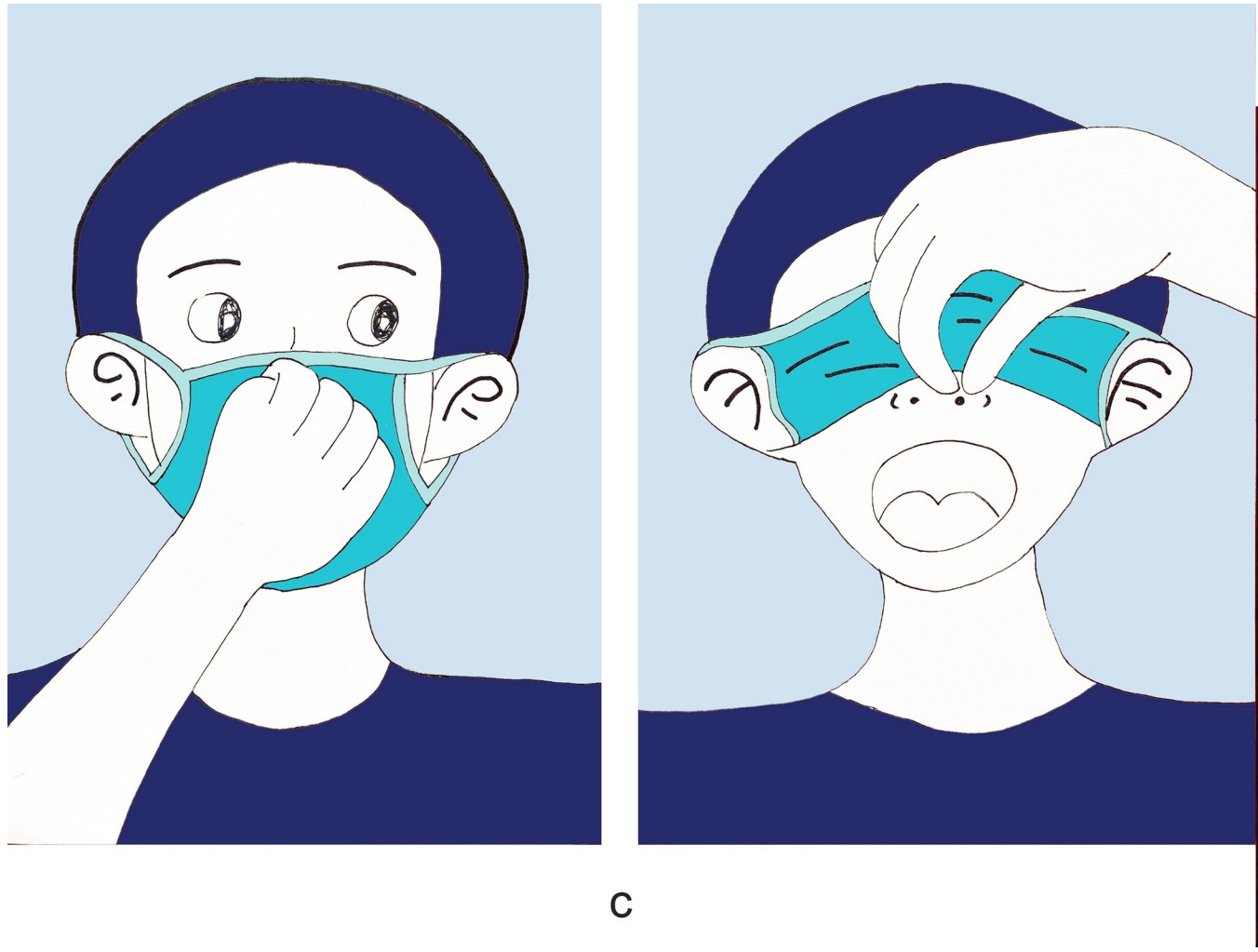

C

**Fig 3. It shows the wrong way of taking off masks.** Touch the surface of the mask, and lift the mask up to expose the mouth. This accounted for 3.7% of participants.

As a result of the highly infectious nature of COVID-19 and the ongoing severity of the global epidemic, face masks are ubiquitous in daily life and communication. We assessed the behavioral differences and correlates of mask usage and found that 73.2% of participants with a high level of health prevention knowledge about mask-wearing during the large-scale nucleic acid sampling site in China, similar to the findings of Tan [9]. We assessed health prevention knowledge and the behavior of face mask use, mainly mask-removal.

Previous studies focused on mask-wearing, especially compliance differences between rural and urban areas [21, 22], and selecting different face masks [23]. In contrast, we focused on one detail (the mask-removal method). We found that only 53.7% of participants removed masks correctly. We found three wrong ways to take off masks; 28.8% of participants used the wrong way in that they touched the mask's surface and lifted it down during nucleic acid sampling (Fig 4). These respondents thought it was a convenient way to remove the mask; 3.7% of participants lifted the mask up to expose the mouth. The wrong method was confusing (Fig 3): Participants indicated that it was a safe way to remove masks; although the mouth was exposed during nucleic acid sampling, the nose was covered.

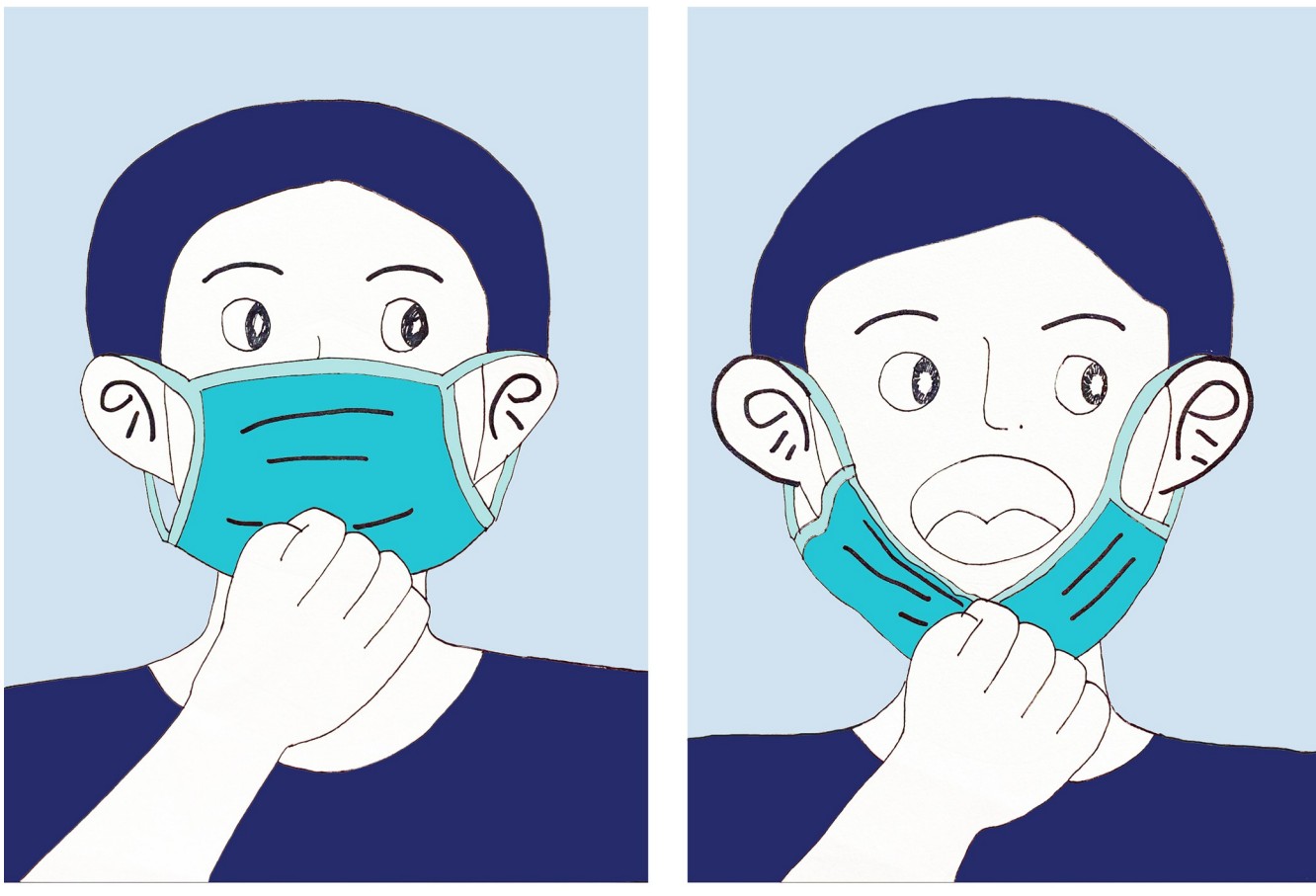

d

**Fig 4. It shows the wrong way of taking off masks.** Touch the surface of the mask, lift the mask down then expose the nose and mouth. This accounted for 28.8% of the participants.

Hand hygiene is a critical factor in preventing health care and is recommended to reduce the spread of COVID-19 and has been reproducibly found to be effective [24]. We focused on the opportunity moment of hand hygiene: We asked, "If you touch the outside of the mask would you perform hand hygiene?"Only 70.3% of participants reported performing hand hygiene after touching the outside of the mask. Therefore, hand hygiene still needs to be emphasized in nucleic acid testing sites, and disinfectants or disinfectant wipes are recommended on them to facilitate hand hygiene.

In addition, we found that health prevention knowledge and free face mask distribution correlated with these details (health prevention behavior of mask-wearing).

**Table 3. Binary logistic regression analysis on the influencing factors of health prevention behaviors of take off mask.**

| Variables | Take off mask | | |
|---|---|---|---|
| | β | Odds ratio (95% CI) | P |
| Good knowledge | 1.91 | 6.75 (4.91, 9.28) | <0.001 |
| Free face mask distribution | 0.38 | 1.46 (1.11, 1.93) | 0.007 |

**Table 4. Binary logistic regression analysis on the influencing factors of health prevention behaviors of hand hygiene.**

| Variables | Hand hygiene | | |
|---|---|---|---|
| | β | Odds ratio (95% CI) | P |
| Good knowledge | 0.88 | 2.42 (1.83, 3.19) | <0.001 |
| Free face mask distribution | 0.74 | 2.10 (1.59,2.78) | <0.001 |

Knowledge promotion is a successful strategy that has altered many health behaviors. A higher level of health literacy is positively associated with higher adherence to the prevention measures against COVID-19 [25]. The influence of health behavior is often the acquisition of knowledge, regardless of educational background. Even simple, brief, and easily conveyable messages can positively impact behaviors [26]. In our survey, participants with better knowledge showed better behavior regarding mask-wearing. Good mask-wearing habits were linked to how much health education about masks was received. This finding supported the hypothesis proposed by Firouzbakht et al. [27]. Knowledge may improve public behaviors and maintain cautious and preventive behavior regarding infectious diseases.

Social media may have a positive effect on health literacy behavior. Netizens have the highest trust in national media network platforms and local government websites [28]. The government should enhance the power and influence of information dissemination in knowledge popularization, news reporting, and information dissemination, carry out social education, and implement COVID-19 prevention and safety measures [29].

Free mask distribution is another factor affecting health prevention behavior. We found that 72.9% of participants reported that the workplace provided masks to employees. Economic factors are crucial. The residents who bought masks out of pocket also demonstrated poor mask-wearing behaviors [30]. Participants might consider it a waste to replace and discard masks, and they would not care about the details of wearing masks correctly. Free mask distribution could encourage good mask-wearing behaviors. The finding supported the hypothesis proposed by Fretheim et al., who suggested that free distribution can increase face mask compliance [31]. Mask distribution can be regarded as social welfare to promote good mask-wearing behavior and combat the COVID-19 epidemic.

The COVID-19 outbreak was a large-scale event with global impacts on health, economics, and politics. This study provides compelling data regarding acceptable face mask use during the acute phase of the pandemic, taking advantage of the RNA mass testing in China to generate a random sample of citizens. The study validates procedures that may be used in future similar scenarios.

However, there are some limitations. First, the participants were drawn from only one city and might not be generalizable to other locations. We did not study the relationship between adverse behavior and the incidence of positive cases of COVID-19. Finally, the behaviors were self-reported, introducing possible reporting bias. Our findings should be regarded with caution.

## Conclusions

This study aimed to investigate the behavior of Chinese residents wearing masks during nucleic acid testing. Knowledge and free mask distribution were associated with correct face mask usage. These findings suggest that it is necessary to disseminate knowledge regarding mask-removal and hand hygiene. Mask distribution policy is linked to the use of masks. Our findings have implications for the ongoing public health emergencies and health prevention behavior during epidemics of infectious diseases spread by respiratory droplets.

## Acknowledgments

We are grateful for the support from ChongChang Zhou for the research design. We thank Qi Zhang, Yue Lv, and HongYi Dong for conducting the surveys. Thanks to WenXia Cheng for creating drawings of how to remove the mask.

## Author Contributions

**Conceptualization:** Jingjing Ma.

**Data curation:** Yiqing Zhang.

**Formal analysis:** Yiqing Zhang.

**Funding acquisition:** Shunshun Lu.

**Investigation:** Shunshun Lu, Shiyong Chen, Yuezheng Rong, Zhengzheng Wang.

**Methodology:** Jingjing Ma, Shunshun Lu, Yuezheng Rong.

**Project administration:** Jingjing Ma.

**Resources:** Shiyong Chen.

**Supervision:** Yuezheng Rong.

**Validation:** Shiyong Chen.

**Visualization:** Zhengzheng Wang.

**Writing – original draft:** Jingjing Ma, Yiqing Zhang.

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
