## [Decision Letter · Decision Letter 0]

21 Nov 2022

PONE-D-22-28370Do you take off your mask correctly? a survey during COVID-19 pandemic in NingBo, ChinaPLOS ONE

Dear Dr. Ma,

Thank you for submitting your manuscript to PLOS ONE. After careful consideration, we feel that it has merit but does not fully meet PLOS ONE’s publication criteria as it currently stands. Therefore, we invite you to submit a revised version of the manuscript that addresses the points raised during the review process.

A minor revision is recommend incorporating a point-to-point authors reply.

We look forward to receiving your revised manuscript.

Kind regards,

Srikanth Umakanthan

Academic Editor

PLOS ONE

Journal Requirements:

Additional Editor Comments:

A minor revision is recommend incorporating a point-to-point authors reply.

Reviewers' comments:

Reviewer's Responses to Questions

**Comments to the Author**

1. Is the manuscript technically sound, and do the data support the conclusions?

Reviewer #1: Yes

Reviewer #2: Yes

2. Has the statistical analysis been performed appropriately and rigorously? 

Reviewer #1: Yes

Reviewer #2: Yes

3. Have the authors made all data underlying the findings in their manuscript fully available?

Reviewer #1: Yes

Reviewer #2: Yes

4. Is the manuscript presented in an intelligible fashion and written in standard English?

Reviewer #1: Yes

Reviewer #2: Yes

5. Review Comments to the Author

Reviewer #1: Abstract: Needs to be curtailed and specific. Include the type of study undertaken for this research.

Introduction and discussion can be strengthened by including the following points:

1. include a sentence on origin, transmission of COVID-19 (refer and cite: doi: 10.1136/postgradmedj-2020-138234)

2. The role of government in combatting COVID-19 in comparison with the other regions (refer and cite: doi: 10.3389/fpubh.2022.844333).

3. The role of social environmental predictors of COVID-19 in general (refer and cite: doi: 10.3390/vaccines10101749)

4. The effect of vaccine hesitancy in COVID-19 and its effects in trust in science (refer and cite: doi: 10.3390/vaccines9101064)

5. The calibre of predictors in COVID-19 (refer and cite: doi: 10.1136/postgradmedj-2021-141365)

Materials and methods: Explain more in detail about the survey, the target population, inclusion and exclusion criteria, if any missed cases were included?

Statistics: divide the covariates into dependent and independent variables.

Mention if bias was generated, and if yes, how was it limited/regressed.

The importance of choosing the statistical relevance should be addressed.

Limitations and strengths need to be included in the end of discussion.

Reviewer #2: The article needs minor revision in the form of grammarly check and also to elaborate the statistical methods more in detail in the section.The introduction, discussion and conclusion should highlight the importance of COVID-19 literature instead of directly focusing on the results of your study.

6. PLOS authors have the option to publish the peer review history of their article (what does this mean?). If published, this will include your full peer review and any attached files.

Reviewer #1: No

Reviewer #2: No

---

## [Author Response · Author response to Decision Letter 0]

28 Nov 2022

I have recovered the modification information in detail in the “Response to Reviewers” .

---

## [Editor Report · Decision Letter 1]

1 Dec 2022

Do you take off your mask correctly? a survey during COVID-19 pandemic in NingBo, China

PONE-D-22-28370R1

Dear Dr. Ma,

We’re pleased to inform you that your manuscript has been judged scientifically suitable for publication and will be formally accepted for publication once it meets all outstanding technical requirements.

Kind regards,

Srikanth Umakanthan

Academic Editor

PLOS ONE
---

## [Editor Report · Acceptance letter]

5 Dec 2022

PONE-D-22-28370R1 

Do you take off your mask correctly? a survey during COVID-19 pandemic in Ningbo, China 

Dear Dr. Ma:

I'm pleased to inform you that your manuscript has been deemed suitable for publication in PLOS ONE. Congratulations! Your manuscript is now with our production department. 

Kind regards, 

on behalf of

Dr. Srikanth Umakanthan 

Academic Editor

PLOS ONE